# The Influence of MSR-B Mg Alloy Surface Preparation on Bonding Properties

**DOI:** 10.3390/ma16103887

**Published:** 2023-05-22

**Authors:** Katarzyna Łyczkowska, Damian Miara, Beata Rams, Janusz Adamiec, Katarzyna Baluch

**Affiliations:** 1Faculty of Materials Engineering, Silesian University of Technology, Krasińskiego 8, 40-019 Katowice, Poland; 2Łukasiewicz Research Network–Institute of Welding, 44-100 Gliwice, Poland; damian.miara@git.lukasiewicz.gov.pl (D.M.); beata.rams@git.lukasiewicz.gov.pl (B.R.)

**Keywords:** adhesive bonding, adhesive layer, surface treatments, magnesium alloy, MSR-B/QE22/(MgAg2RE2Zr), one-component epoxy

## Abstract

Nowadays, industrial adhesives are replacing conventional bonding methods in many industries, including the automotive, aviation, and power industries, among others. The continuous development of joining technology has promoted adhesive bonding as one of the basic methods of joining metal materials. This article presents the influence of surface preparation of magnesium alloys on the strength properties of a single-lap adhesive joint using a one-component epoxy adhesive. The samples were subjected to shear strength tests and metallographic observations. The lowest properties of the adhesive joint were obtained on samples degreased with isopropyl alcohol. The lack of surface treatment before joining led to destruction by adhesive and mixed mechanisms. Higher properties were obtained for samples ground with sandpaper. The depressions created as a result of grinding increased the contact area of the adhesive with the magnesium alloys. The highest properties were noticed for samples after sandblasting. This proved that the development of the surface layer and the formation of larger grooves increased both the shear strength and the resistance of the adhesive bonding to fracture toughness. It was found that the method of surface preparation had a significant influence on the resulting failure mechanism, and the adhesive bonding of the casting of magnesium alloy QE22 can be used successfully.

## 1. Introduction

Magnesium (Mg) alloys were first developed in the 1940s, and due to World War II, they were used primarily for military applications (e.g., B36 and B52 bombers). After the war, Mg alloys were also employed in civil production [1,2,3,4,5].

The advantages of Mg alloys have attracted the automotive industry, which, over the years, utilized them in car parts production, e.g., engine block housings, seat frames, etc. The development of Mg alloys focuses on the reduction of the weight of certain elements while maintaining or improving their existing properties. This includes increasing the material’s plasticity, creep resistance, and stiffness, among other properties. The process of modifying the properties of Mg alloys depends on selecting the appropriate chemical composition or manufacturing technology (e.g., pressure casting) [6,7,8,9,10,11,12,13].

Mg alloys are characterized by a high strength and a low density of approximately 1.80 g/cm^3^ (the density of aluminum alloys is approximately 3 g/cm^3^, and that of carbon steel is approximately 7.80 g/cm^3^). This leads to the weight reduction of the car’s structure by up to 15%, which in turn influences the reduction of exhaust emissions and fuel economy. Additionally, Mg alloys are easy to use in repair conditions [14,15,16,17,18,19,20,21,22,23,24,25].

Currently, due to the development of manufacturing technology, plastic deformation, and heat treatment, Mg alloys are widely used not only in the automotive, aviation, and energy industries but also in electronics, medicine, textiles, etc. [26,27,28,29,30].

One Mg alloy group is the Mg-Ag-RE-Zr group, which exhibits a high yield point, proper tensile strength, and fatigue resistance up to 300 °C. These properties are a result of the presence of rare earth elements, which create stable intermetallic phases at the boundary grains. Furthermore, this limits the grain boundary sliding effect in the microstructure of the material.

The Mg-Ag-RE-Zr group is characterized by good casting properties and machining, which contribute to its application in complex-shaped element casting. However, the disadvantages of these alloys lay in their tendency to crack and deform during heat treatment, their high cost due to the presence of silver, and their low corrosion resistance. These alloys are mainly used in the aviation industry for chassis wheels, engine bodies, gearbox housings, and helicopter rotor heads. Moreover, they are employed in the car industry (the main recipient is Rolls Royce) and the military industry. The chemical composition of the Mg-Ag-RE-Zr group is presented in Table 1 [18,31,32,33,34].

The high demand for new construction and technological solutions encourages engineers to develop novel materials and examine the joining technologies. This trend is related to the development of adhesive technology, which is an ancient approach but has evolved over the last 80 years, resulting in the expansion of plastics [35,36].

The advantages of bonding employed to join metal materials include the ability to combine different materials (in physical and chemical properties), limit the impact of the joints on the properties of the base material, and combine elements of various dimensions and shapes. Additional advantages are sealing the joining point, protecting against moisture, eliminating additional surface treatment after the process, and lowering the structure mass or even uniform distribution of loading in the joining point [37,38,39,40,41]. It is also important that the adhesive not only meets the production requirements but the requirements of aging stability and mechanical properties. According to ISO 9001 [42], adhesive bonding belongs to the ‘special process’ group, similar to welding, riveting, twisting, and soldering techniques. For these processes, the results cannot be determined by non-destructive testing [43].

Adhesive bonding has been widely described and defined in the literature. According to DIN EN 923 [35,41,42,44,45,46], an adhesive is a nonmetallic substance capable of joining materials by surface bonding (adhesion), with the bond possessing adequate internal strength (cohesion). Another popular theory of bonding is the Kinloch theory [47], which defines an adhesive as a material that, after its application to the material’s surfaces, leads to their bonding and prevents separation. Adams et al. [48] showed that a structural adhesive is an adhesive that influences the stiffness and strength of the structure and can withstand significant loads. Other reports have also described adhesive bonding, e.g., Dostal [49], Tong L, Steven G.P. [50], Petrie E.M. [51], Packham [52], Possart [53], Lacombe [54], Ebnesajjad S (ed) [55], and da Silva and Öchsner [56]. Another important aspect of adhesive bonding technology is the correct design of the adhesive joint. To reduce stresses in the joining point and due to the size of the joining surface, it is recommended to construct lap joints for the joints exposed to shear. However, increasing the length of the adhesive joint (the overlap) can lead to increased joint strength. Exceeding the length limit can promote faster destruction of the joint [57].

One of the most important steps during the bonding process is surface preparation [47,54,58,59,60,61]. Its purpose is to remove the external layers (e.g., oils, greases), the adsorption layers (e.g., water molecules), and the reactive layers (e.g., oxides, hydroxides). Appropriate surface preparation increases the specific surface area of the joining point. This increases the strength of the joint by increasing the adhesion between the material and the adhesive. The surface treatment is performed until the base material is exposed, which allows the oxide layer to be recreated under controlled conditions [62,63,64].

The method of surface preparation and the selection of bonding parameters determines the type of destruction of the adhesive joints. Cohesive cracking is one of the most desirable approaches to ensure that the material is properly prepared before the process. The basic types of damage to adhesive joints are shown in Figure 1 [65].

The current knowledge about surface preparation before adhesive bonding mainly concerns composites reinforced with carbon fiber [66] and aluminum alloys [67]. The surface preparation of Mg alloys for bonding was previously discussed by Ren 34 DX, Liu LM, Li YF [68], and Xu W, Liu L, Zhou Y, Mori H, Chen DL [69,70]. It was found that bonding can be successfully used as a joining method due to the joint’s high strength in shear conditions. However, a proper preparation method for bonding is a vital requirement. Mirski et al. described the influence of surface preparation on the properties of adhesive joints [71]. They revealed that light metal alloy (AZ31B, Aluminum 5754, Titan Grade 2) surface preparation by abrasive blasting led to 25% higher joint strength in comparison with grinding. Tang 118 et al. [72] described the influences of the duty cycle on the bonding strength of the AZ31B magnesium alloy by microarc oxidation treatment. The authors proved that the microarc oxidation significantly improved the adhesive bonding of the AZ31B magnesium alloy. As the duty cycle on film porosity increased, the lap shear strength of the bonding joints increased. The reason was attributed to the larger porosity and enhanced mechanical interlocking effect. Equally satisfactory results of the effects of phosphate pretreatment were described by Yuea, et al. [73]. The authors showed that it can produce a significant improvement in the corrosion resistance and adhesive bond performance of phosphate by means of increasing the polarization resistance in NaCl solution.

Among the known methods of the surface preparation of magnesium alloys, reports have employed degreasing, mechanical treatment, and chemical and electrochemical treatment. Mechanical examinations have shown enhanced properties for joints prepared by abrasive blasting (sandblasting) compared with etching or grinding. Equally promising results were presented for chemical pre-treatment. Literature analysis indicated that the obtainment of the required strength of the adhesive joint is associated with both the selection of the appropriate adhesive and the preparation of the bonding surface. Numerous data have been published regarding the bonding of light metals, such as aluminum, titanium, and their alloys. However, detailed information examining the bonding of Mg alloys is rather limited, which was the motivation for the authors to publish their own results.

## 2. Materials and Methods

### 2.1. Testing Methodology

The specimens were prepared from the cast Mg alloy MSR-B (QE22). Materials were cut into slices of two different thicknesses (Table 2). MSR-B samples were used to prepare single-lap adhesive joints. The geometry of Variant No. I samples (25 × 100 × 1.6; lap 12.5) was prepared according to the EN 1465 standard [74]. Only cohesion failure mechanisms in Mg alloys were observed. Subsequently, samples with dimensions of 25 × 100 × 3.0; lap 12.5 mm (Variant No. I.I.) were prepared. Due to the same failure mechanism as in Variant No. I (cohesion failure), the results are not described in the manuscript. The lap geometry changes in Variant No. II led to various failure mechanisms, e.g., adhesion. This provided an opportunity to evaluate the influence of surface preparation on the properties of the adhesive joint. For each of the variants, 18 samples were prepared (sets of 6 for each method of surface preparation before adhesive bonding). The scheme of the tested adhesive joints is shown in Figure 2.

### 2.2. Surface Preparation

Three different methods were used for surface preparation before bonding, which was performed on two different thicknesses (Table 2). The methods of surface preparation on MSR-B Mg alloys are presented in Table 3.

### 2.3. Roughness Test

Roughness measurements for each surface preparation method were performed by Technolutions on a HIROX NPS profilometer (Tokyo, Japan). Roughness values are displayed as the mean values Ra and Rz of the surfaces prepared by different methods.

### 2.4. Adhesive Bonding

One-component thermosetting epoxy adhesive LOCTITE^®^ EA 9514 was selected as an adhesive for the presented research. It was characterized by chemical resistance and high mechanical performance (shear resistance, cleavage strength). The epoxy adhesive was applied on the cleaned surfaces and then cured in an oven at 175 °C for 90 min.

### 2.5. Adhesion Strength

The lap joints were tested under the shear conditions. The adhesion was determined by the tensile shear test according to ISO 4587 ‘Adhesive Lap-Shear Strength of Rigid-to-Rigid Bonded Assemblies’ [75]. The tensile shear tests of the single-lap adhesively bonded joints were examined using an INSTRON 4210 testing machine (Norwood, MA, USA) and a 5 mm/min speed rate.

### 2.6. Metallographic Tests

The microstructure of surfaces obtained as a result of three different types of preparation for bonding were examined under a scanning electron microscope (SEM) JEOL JCM-6000 Neoscope II (Tokyo, Japan). The investigations were conducted using the Secondary Electron (SE) and Backscattered Electron (BSE) techniques at magnifications of up to 1000×.

The cross-section of the adhesive joint was prepared by cold inclusion in epoxy resin, then grinding on abrasive papers with gradients of 120, 320, 500, 1200, and 2500 μm and polishing successively on pastes with grain sizes of 3, 1, and 0.25 μm. The metallographic examinations were conducted using an Olympus GX71 (Tokyo, Japan) light microscope (LM) at magnifications of up to 500×.

## 3. Results and Discussion

Bonding is a method for combining metal materials that is used with increasing success in many industries. The obtainment of a high-quality adhesive joint and the repeatability of parameters is related to, among other factors, the proper preparation of the surface for bonding. Herein, our study investigated single-lap adhesive joints made on MSR-B Mg alloy whose surfaces were prepared for bonding using three different methods (Table 3) and varied in thickness (Table 2). Adhesive joints were examined under shear conditions followed by metallographic observations.

### 3.1. Roughness Measurements

The roughness tests of the samples after degreasing with isopropyl alcohol were characterized by the lowest roughness. The average Ra and Rz values were ±0.4 μm and ±4.2 μm, respectively. Sample surfaces treated with grinding using sandpaper and additionally degreased showed increased Ra and Rz values of ±1.5 μm (1.28–1.51 μm) and ±12 μm (10.44–13.94 μm), respectively. A significant increase in bonding strength was obtained for samples treated with abrasive blasting. The Ra and Rz values were ±3 μm (2.77–3.28 μm) and ±28 μm (25.37–32.29 μm), respectively. A profilograph measurement of the Mg alloy surface after implementing the Gaussian filter conforming to ISO 13565-1 [76] is shown in Figure 3.

### 3.2. Metallographic Observation of Pre-Treatment Surface

Microstructure analysis was performed on the surfaces prepared for bonding and revealed significant differences in the morphology of the surfaces depending on the preparation method. The area of contact of the base material with the adhesive was irregular, which was due to material cutting before pre-treatment (Figure 4a,d). The preparation method using sandpaper led to the elimination of impurities from the material’s surface. The scratches and furrows with irregular shapes and a width of around 100 μm were visible on the material’s surface (Figure 4b,e). The surface after the abrasive blasting technique with corundum particles revealed the most ununiform surface between the compared preparation methods The significant influence of intense scratches of the corundum particle impact marked the surface geometry (Figure 4c,f).

The Mg alloy surfaces, which were prepared for bonding by degreasing with isopropyl alcohol, were characterized by a relatively even and smooth surface with small scratches (Figure 5a). After grinding with sandpaper, the Mg alloy had a rough surface. The surface was visibly developed, with visible longitudinal, oriented scratches and sharp edges (Figure 5b).

The surface of the material after blast-abrasive treatment exhibited numerous irregularities. These irregularities were observed along the entire length of the joint (Figure 5c), which, due to the preparation conditions, revealed a positive effect on the development of the contact surface compared with that cleaned with isopropyl alcohol.

The results of the shear test for joints with dimensions of 25 × 100 × 3 mm and an overlap of 6 mm (Variant No. II) are presented in Figure 6. Joints with surfaces prepared by cleaning with isopropyl alcohol possessed a shear strength of 20.09–22.3 MPa. Those with a surface ground with 120 μm sandpaper revealed a higher shear strength of 26.3–27.7 MPa. This was related to the changes in surface geometry under the influence of large grinding particles during preparation.

However, the highest properties were obtained for Mg alloys with surfaces prepared for bonding by abrasive blasting with alumina oxide. The shear strength was 35.7–36.8 MPa. Hence, this showed the influence of the surface preparation on the bonding quality of the MSR-B Mg alloy. The high mechanical properties of the joints were related to the removal of the external and absorption layers in base material before bonding, e.g., oxides and impurities and the development of the material surface layer.

The mechanical investigations were performed on samples subjected to all types of surface preparation. The results showed that the shear test for samples with 25 × 100 × 1.6 mm dimensions and a 12.5 mm overlap joint (Variant No. I) had a ruptured base material, regardless of the preparation method employed (Figure 7a–c). For each of the tested joints, the breakage place occurred near the adhesive joint. This area was the weakest point of the construction of the entire joint. The same type of failure was observed for samples with 25 × 100 × 3.0; 12.5 mm of lap dimensions (Variant No. II).

Additionally, this type of joint failure indicated that, due to the high properties of the tested adhesive, they could be employed in the industry for the structural bonding of Mg alloys. In the case of MSR-B Mg alloy surfaces (Variant No. II) degreased with isopropyl alcohol, cohesion and adhesion were observed as degradation mechanisms. The presence of different mechanisms may have been the result of insufficient preparation of the surface for bonding. The surfaces ground with 120 μm sandpaper had a different mechanism of degradation. However, the ground area of the joint was destructed by cohesion. Therefore, the surface after grinding was better prepared than that cleaned with isopropyl alcohol. In the case of the surface prepared by abrasive blasting, the adhesive joint was degraded by a cohesive mechanism. This was caused by a decrease in the strength of the intermolecular bonds of the adhesive as a result of the external load increasing. The observations of the fracture topography after bonding of MSR-B Mg alloys with 25 × 100 × 3 mm dimensions and a 6 mm overlap (Variant No. II) are shown in Figure 7d–f.

Subsequently, the surfaces after the cracking were observed on a scanning electron microscope. On the surface of materials degreasing with isopropyl alcohol, an adhesive cracking mechanism was observed among others, which was indicated by smooth areas on the magnesium alloy samples (Figure 8a,d). At the failure border, there were places where a small amount of adhesive residue.

Single reinforcing particles and numerous delaminations in adhesive layer were formed as a result of cracking. On the material prepared by grinding with sandpaper, numerous scratches filled with adhesive were revealed (Figure 8b,e). The largest areas of remaining adhesive were observed in grooves of irregular shape and depth. It is a proof for good adhesive penetration into the material, which significantly increases the size of contact surface, and thus increases the strength of the adhesive joint compared to surfaces cleaned only with alcohol.

On the samples prepared by sandblasting, a cohesive failure mechanism was observed (Figure 8c,f). The crack occurred in the layer of adhesive. On the entire surface of the adhesive fracture, evenly distributed air bubbles and gas pores were observed, which are characteristic of epoxy adhesives.

On the basis of the test results, it was found that the materials prepared by degreasing with alcohol were characterized by the lowest values of shearing strength and the lowest values of surface roughness. The increase in adhesive joint strength was observed for materials ground with sandpaper. The best mechanical properties were observed for materials after sandblasting. It has been proven that the development of the surface area and the formation of larger grooves increase both the shear strength and the resistance of the adhesive joint to cracking, and similar results are presented in this paper [66,67,71,72,73]. The authors proved that the quality of surface preparation influenced the mechanical properties of the joining point. On the basis of the preliminary results of the research, it was found that the presented technology can be used successfully for MSR-B magnesium alloy bonding.

## 4. Conclusions

On the basis of the conducted research and analysis of the obtained results, the following conclusions were drawn:-The increased strength of the adhesive joint was related to the even development of the adhesive surface and the elimination of impurities.-Blast-abrasive treatment as surface preparation for the bonding of MSR-B Mg alloy led to the highest shear strength of the adhesive joint (an increase of 31% compared with grinding and 64% compared with degreasing with isopropyl alcohol).-The application of LOCTITE^®^ EA 9514 one-part thermosetting epoxy adhesive for bonding materials 1.6 mm in thickness resulted in the destruction of the base material, not the adhesive. This showed that the high properties of the adhesive could allow it to be employed in industries for the bonding of Mg alloys.

## Figures and Tables

**Figure 1 materials-16-03887-f001:**
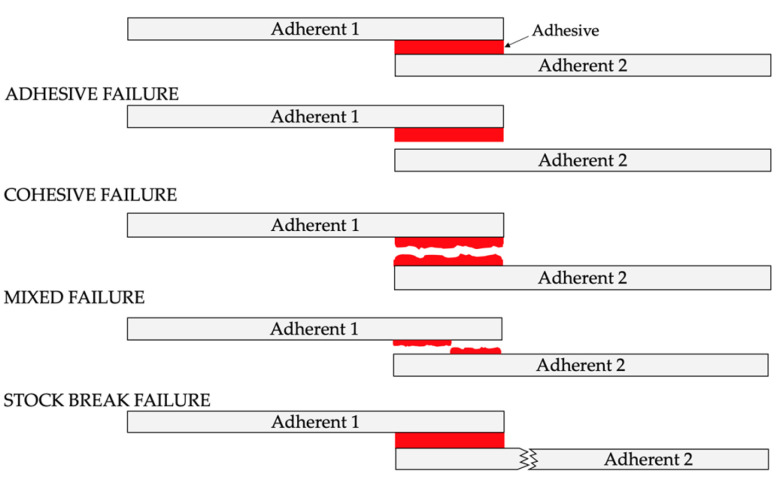
Examples of cohesive and adhesive failures [65].

**Figure 2 materials-16-03887-f002:**
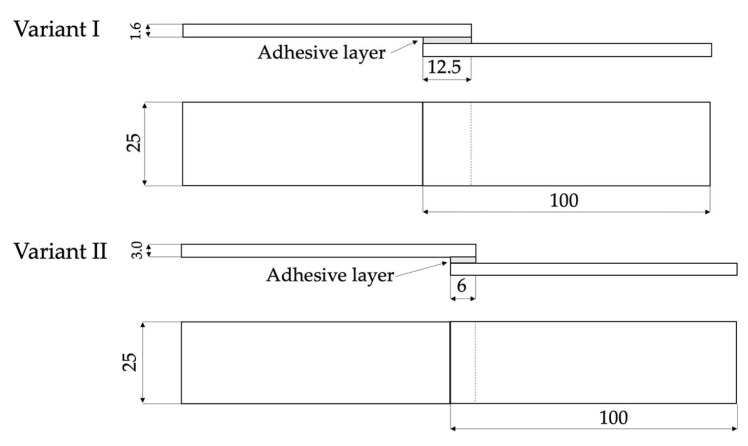
The scheme of joint used in the static shear test, mm.

**Figure 3 materials-16-03887-f003:**
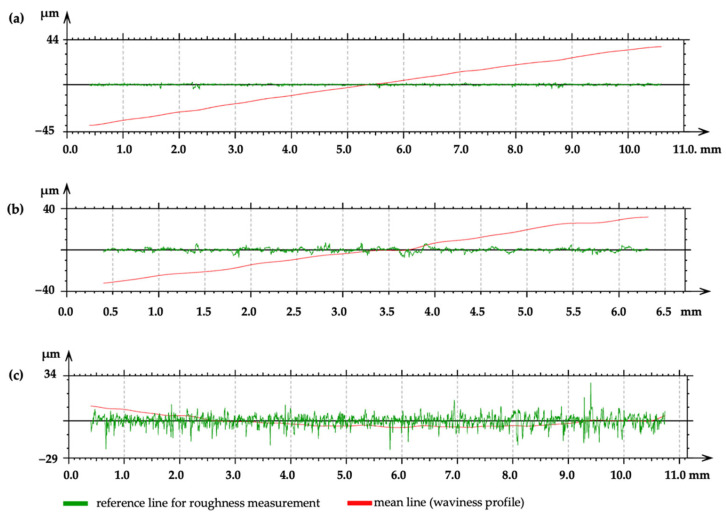
The profilograph measurements of Mg alloy surface after cleaning (**a**), grinding (**b**), and abrasive blasting (**c**).

**Figure 4 materials-16-03887-f004:**
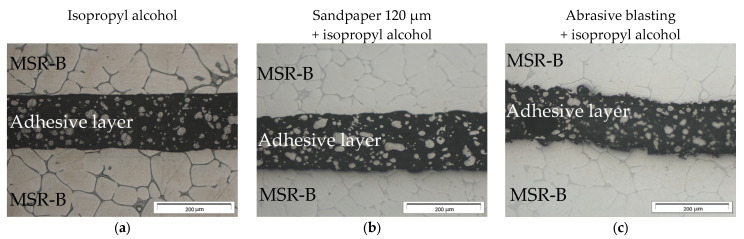
Microstructure of the adhesively bonded joints of MSR-B Mg alloy samples after pre-treatment with (**a**,**d**) isopropyl alcohol, (**b**,**e**) grinding with sandpaper 120 μm, or (**c**,**f**) abrasive blasting.

**Figure 5 materials-16-03887-f005:**
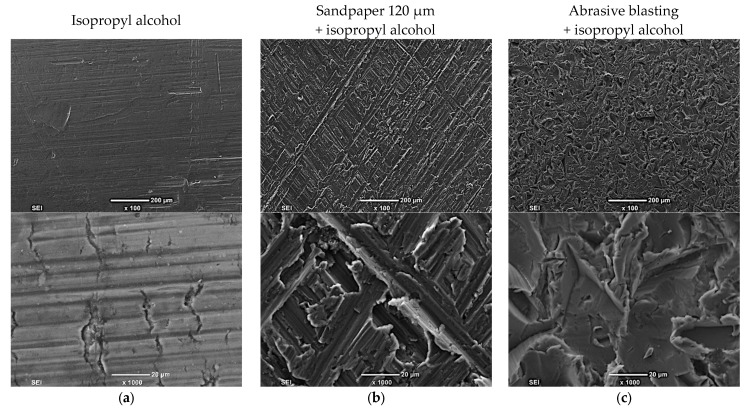
The macrostructure and morphologies of MSR-B alloy samples after pre-treatment with (**a**) isopropyl alcohol, (**b**) grinding with 120 μm, or (**c**) abrasive blasting with corundum.

**Figure 6 materials-16-03887-f006:**
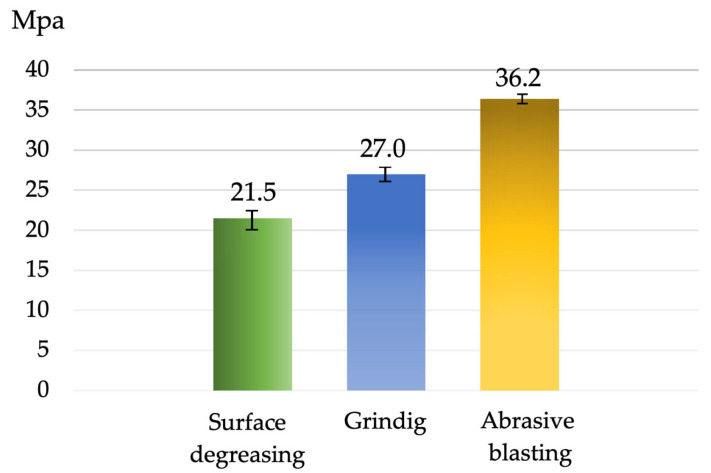
Shear strength of the adhesive-bonded joints in relation to pre-adhesive bonding surface preparation (sample dimension of 25 × 100 × 3 mm).

**Figure 7 materials-16-03887-f007:**
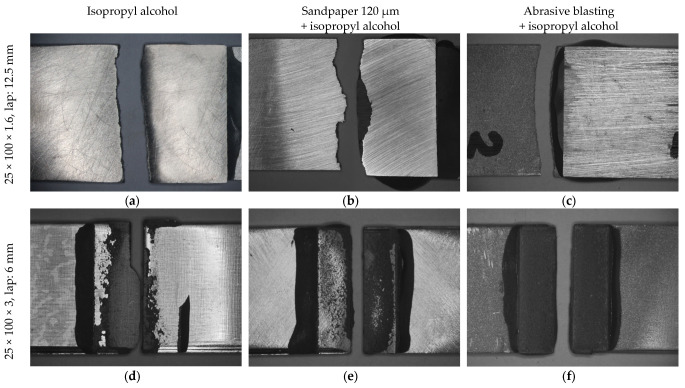
Effect of surface treatment on the fractography of the adhesive-bonded lap shear. (**a**–**c**) samples with 25 × 100 × 1.6; 12.5 mm, (**d**–**f**) samples with 25 × 100 × 3.0; 6.0 mm.

**Figure 8 materials-16-03887-f008:**
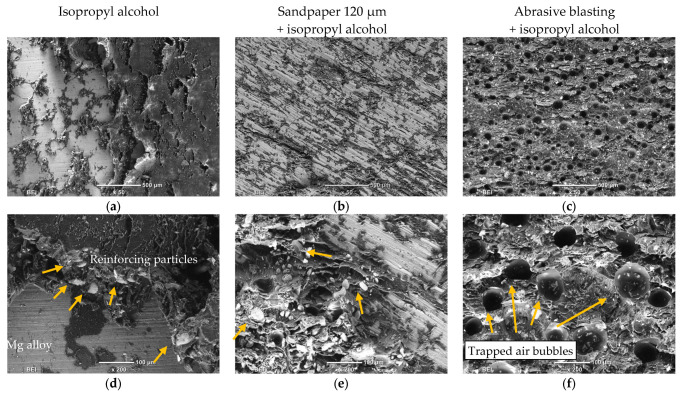
The surfaces of the samples after treatment, SEM. (**a**,**d**) visible delaminations in adhesive layer with reinforcing particles, (**b**,**e**) numerous scratches filled with adhesive and numerous reinforcing particles, (**c**,**f**) air bubbles and gas pores in adhesive layer after sandblasting.

**Table 1 materials-16-03887-t001:** Chemical composition of Mg alloys from Mg-Ag-RE-Zr group.

Alloy	Chemical Composition, [%]			
	Mg	Ag	RE	Zr
EQ21	Balance	1.5	2.1	0.7
QE22 (MSR-B)	Balance	2.0 ÷ 3.0	2.0 ÷ 3.0	0.4 ÷ 1.0

RE—a mixture of rare earth elements containing 85% by mass neodymium and 15% praseodymium (didymium).

**Table 2 materials-16-03887-t002:** Fabrication of adhesive-bonded lap.

Variant No.	Sample Dimensions, mm	Adhesive Length of Layer, mm
I	25 × 100 × 1.6	12.5
II	25 × 100 × 3.0	6.0

**Table 3 materials-16-03887-t003:** Methods and parameters of the surface treatments.

No.	Pre-Treatment	Detail
1	Surface degreasing	Cleaning with isopropyl alcohol
2	Grinding	Grinding with 120 μm sandpaper until unpolished (matt), cleaning with isopropyl alcohol
3	Abrasive blasting	Blasting with 120–150 μm corundum under 0.4 MPa pressure, cleaning with isopropyl alcohol

## Data Availability

The data supporting reported results is not stored in any publicly archived datasets. The readers can contact the corresponding author for any further clarification of the results obtained.

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
