# Peer review of "The Influence of MSR-B Mg Alloy Surface Preparation on Bonding Properties"

_materials, 2023, doi:10.3390/ma16103887_

Round 1

Reviewer 1 Report

The manuscript presents an experimental investigation of the lap-shear strength of the bonded joint of Mg alloy. A couple of comments are needed to be addressed to enhance the discussion of the work:

1- The roughness measurement for all the samples should be reported to better describe the strengthening modes because it was proved with large grooves at the surface, the fracture toughness of the joint increases and thus the shear strength should increase [1].

[1] Improving mode II fracture toughness of secondary bonded joints using laser patterning of adherends. Composites Part A: Applied Science and Manufacturing, 134, 105892

2- SEM for the fracture surface will add great value to understanding the strengthening mechanisms.

Author Response

Dear Reviewer,  

Thank you for your comments. We are grateful for taking your time to read our paper and constructive comments. We have reviewed the comments and have revised the manuscript accordingly.

We hope the revised version is suitable for publication.

Your sincerely,

Katarzyna Łyczkowska

Point 1: The roughness measurement for all the samples should be reported to better describe the strengthening modes because it was proved with large grooves at the surface, the fracture toughness of the joint increases and thus the shear strength should increase [1].
Response 1: The roughness measurement were reported in the manuscript. Thank you for the suggestion.

Point 2: [1] Improving mode II fracture toughness of secondary bonded joints using laser patterning of adherends. Composites Part A: Applied Science and Manufacturing, 134, 105892
Response 2: Thank you for this article. Cited in the manuscript

Point 3: 2- SEM for the fracture surface will add great value to understanding the strengthening mechanisms.
Response 3: Added in the manuscript. Thank you for the suggestion.

Reviewer 2 Report

Suggiestions are included in the attached document. 

Author Response

Dear Reviewer,

Thank you for your comments. We are grateful for taking your time to read our paper and constructive comments. We have reviewed the comments and have revised the manuscript accordingly.
We hope the revised version is suitable for publication.

Your sincerely,
Katarzyna Łyczkowska

Point 1:  15-16 The following sentence “It has been that the strength of an adhesive joint depends on the adhesive properties between the material and the adhesive, and the adhesive layer's cohesive properties” should be rewritten. Please, add the most relevant result obtained in the work.
Response 1: That was rewritten in the manuscript. Thank you for the suggestion.

Point 2:  Table 1. The authors should also report the Young’s modulus of the materials.
Response 2: Corrected in the manuscript. Thank you for the suggestion.

Point 3: When the authors are reporting some references, the relative number adopted in the bibliography should be reported (e.g. Ren 126 DX, Liu LM, Li YF [69] and Xu W, Liu L, Zhou Y, Mori H, Chen DL in [70] and [71.) (The authors Mirski et al. also described)
Response 3: Corrected in the manuscript. Thank you for the suggestion.

Point 4: The introduction section is mainly reporting insight about adhesive joint. However a work about the surface preparation of Mg alloy substrates should mainly deal with the adopted surface treatment used for Mg alloy substrates. Further, the outcomes of the works should be reported in an extensive way (main result) to give significance to the present work and to comment your result in relation to the ones already published in the literature.
Response 4: Your suggestion were taken into account in the revised version. Thank you very much.

Point 5:  Why the authors decided to test two different sample sizes? This should be reported in the paper.
Response 5: For the materials prepared according to the EN 1465 standard, only cohesion failure mechanism were shown in area of Mg alloys. The changes in sample geometry (increasing the thickness of the material and reducing the length of the overlap) led to adhesive failure between the Mg alloy and the adhesive. This made it possible to evaluate the method of substrate preparing on the quality of the adhesive bonding. This information has been added to the manuscript.

Point 6: 6- The reference [74] is not reported in the bibliography.
Response 6: Corrected in the manuscript. Thank you for the suggestion.

Point 7: think that Olympus is Japanese (Olympus GX71 (Warsaw, Poland)).
Response 7: Corrected in the manuscript. Thank you for the suggestion.

Point 8: Figure 7. A legend for the green and red curves must be reported.
Response 8: Corrected in the manuscript. Thank you for the suggestion.

Point 9: 210-211. What is the source of the small scratches?
Response 9: The small scratches were created during casting cutting. Due to the lack of pre-treatment by grinding or sandblasting, the scratches were not removed from the surface.

Point 10: Figure 9. Report the error bar in the graph (also a 2D graph can be used in this case). Replace the comma with the dot.
Response 10: Corrected in the manuscript. Thank you for the suggestion.

Point 11: The number of replications should be reported in the material section.
Response 11: Corrected in the manuscript. Thank you for the suggestion.

Point 12: Representative curves for the samples 25x100x3 and 25x100x1.6 should be reported.
Response 12: Only the 25x100x3 samples were shown as the only cases where the failure occurred in the adhesive bonding. In the case of 1.6 mm thickness, the crack occurred in the material and had no effect on the strength of the adhesive joint.

Point 13: Section 3.2 the results should be properly commented and compared to what have been presented in the literature.
Response 13: Your suggestion were taken into account in revised version. Thank you very much.

Point 14: Figure 10. It is not clear to me why the authors inserted the study on the sample “25x100x1.6, lap: 12.5 mm” since substrate failures were obtained in all the cases and the strength of the adhesive is lower than the strength of the substrates. Please explain.
Point 14: In the mentioned samples, the crack propagation occurred near to the adhesive joint. It was caused by uniform distribution of shear stresses. Area near to joint border was probably the point of maximal stresses accumulation. It led to brittle and immediate cracking of the material. As a result, the adhesive bond revealed higher properties than the substrate.
However, to clarify, authors decided to remove the mechanical properties from the Tab.1. Thank you for your comments.

Point 15: Figure 11 reports different micrography of the samples. The authors should better work on Figure 11. Honestly, I cannot see significant differences. What is the difference between the white and black parts? Maybe, the authors can work on the figures (use of arrow and signs) to better explain what is shown.
Response 15: Descriptions and lines were added to the manuscript. Thank you for the suggestion.

Reviewer 3 Report

The present work entitled "The influence of MSR-B Mg alloy surface preparation on the bonding properties" considers the study of the properties of glue bonding between Mg alloys.

 This topic is interesting and worthy to be investigated.

The paper is generally well written, with a rather informative introduction.

The methodological part is described in detail. The experiments were well designed. The references are appropriate.

Nevertheless, there are several comments to the authors:

The abstract does not reflect the main results of the article and should be rewritten.

Line 37-38: Non-obvious statement, it is not clear what the authors meant.

Line 98: There should be a reference, not a figure.

Figure 2: What does lgr mean?

Line 127: [71]

Line 132: Titan, not Tytan

The review contains a lot of information separately about magnesium alloys and adhesive joints. Information about the work on gluing magnesium alloys has not been analyzed. It is unclear how the work of the authors differs from the already published works.

(for example, Tang, Yuming, et al. "The influences of duty cycle on the bonding strength of AZ31B magnesium alloy by microarc oxidation treatment." Surface and Coatings Technology 205.6 (2010): 1789-1792.

Liu, Zhong-xia, et al. "Effects of phosphate pretreatment and hot-humid environmental exposure on static strength of adhesive-bonded magnesium AZ31 sheets." Surface and Coatings Technology 206.16 (2012): 3517-3525.)

There is no justification for choosing the length of the overlap, the thickness of the adhesive and the thickness of Mg plates.

Table 2: What does "*" mean?

Table 3: What does "until tarnished" mean?

Why was Cleaning with isopropyl alcohol not used after abrasive blasting?

Line 165: (Tokyo, Japan)

Line 168: Missing source reference 74

Line 178: ISO ISO

Line 190: Olympus GX71 (Warsaw, Poland) ???

Line 193-199: Repetition of already stated information.

Figure 7. Why are profilograms for samples No. 1 and 3 not shown.

Line 212-213: What kind of impurities are we talking about? They are not visible in Fig. 8. What is the non-invasive cleaning method?

Line 214: sandpaper with a gradation of 120 μm?

Figure 9: Why are the results given only for the 25x100x3 samples?

It is not clear why samples of 1.6 mm thickness with a different overlap were used? Why weren't 3mm samples with 12.5mm overlap used?

Line 259-262: What does irregularities on the surface mean?

Figure 11: Missing pictures g, h, i.

Manuscript is recommended to accept for publication after minor revision.

Author Response

Dear Reviewer,

Thank you for your comments. We are grateful for taking your time to read our paper and constructive comments. We have reviewed the comments and have revised the manuscript accordingly.
We hope the revised version is suitable for publication.

Your sincerely,
Katarzyna Łyczkowska

Point 1:
The abstract does not reflect the main results of the article and should be rewritten.
Response 1: The abstract in revised version was rewritten according to your suggestion about more detailed data. Thank you.

Point 2: Line 37-38: Non-obvious statement, it is not clear what the authors meant. 
Response 2: Mg alloys applied on the aircraft external structure were often required repaired. Thanks to their properties, they could be easily repair by welding techniques.

Point 3: Line 98: There should be a reference, not a figure.
Response 3: Corrected in the manuscript. Thank you for the suggestion.

Point 4: Figure 2: What does lgr mean?
Response 4: This figure was deleted from the manuscript.

Point 5: Line 127: [71]
Response 5: Corrected in the manuscript. Thank you for the suggestion.

Point 6: Line 132: Titan, not Tytan
Response 6: Corrected in the manuscript. Thank you for the suggestion.

Point 7: The review contains a lot of information separately about magnesium alloys and adhesive joints. Information about the work on gluing magnesium alloys has not been analyzed. It is unclear how the work of the authors differs from the already published works.
(for example, Tang, Yuming, et al. "The influences of duty cycle on the bonding strength of AZ31B magnesium alloy by microarc oxidation treatment." Surface and Coatings Technology 205.6 (2010): 1789-1792.

Liu, Zhong-xia, et al. "Effects of phosphate pretreatment and hot-humid environmental exposure on static strength of adhesive-bonded magnesium AZ31 sheets." Surface and Coatings Technology 206.16 (2012): 3517-3525.)
There is no justification for choosing the length of the overlap, the thickness of the adhesive and the thickness of Mg plates.
Response 7: The obtained results were compared with the literature data. It is presented in results and discussion section.
The determination of the bonding joint geometry changes are presented in Materials and Methods section.
Thank you for your comments.

Point 8: Table 2: What does "*" mean?
Response 8: This sign was deleted from the manuscript. Thank you for the suggestion.

Point 9: Table 3: What does "until tarnished" mean?
Response 9: Corrected in the manuscript. Now: until unpolished (matt)

Point 10: Why was Cleaning with isopropyl alcohol not used after abrasive blasting?
Response 10: Cleaning with isopropyl alcohol was used after abrasive blasting. Corrected in the manuscript. Now

Point 11: Line 165: (Tokyo, Japan)
Response 11: Corrected in the manuscript. Thank you for the suggestion.

Point 12: Line 168: Missing source reference 74 
Response 12: Corrected in the manuscript. Thank you for the suggestion.

Point 13: Line 178: ISO ISO
Response 13: Corrected in the manuscript. Thank you for the suggestion.

Point 14: Line 190: Olympus GX71 (Warsaw, Poland) ???
Response 14: Corrected in the manuscript. Thank you for the suggestion.

Point 15: Line 193-199: Repetition of already stated information.
Response 15: Corrected in the manuscript. Thank you for the suggestion.

Point 16: Figure 7. Why are profilograms for samples No. 1 and 3 not shown.
Response 18: The profilograms for samples No. 1 and 3 were shown in manuscript.

Point 17: Line 212-213: What kind of impurities are we talking about? They are not visible in Fig. 8. What is the non-invasive cleaning method?
Response 17: This sentence was deleted from the manuscript. Thank you for the suggestion. It is a non-invasive cleaning method means not affect the material surface during pretreatment.

Point 18: Line 214: sandpaper with a gradation of 120 μm?
Response 18: Corrected in the manuscript. Thank you for the suggestion.

Point 19: Figure 9: Why are the results given only for the 25x100x3 samples?
Response 19: Only for samples 25x100x3 the failure occurred in the adhesive layer. In the case of 1.6 mm thickness, the crack occurred in the material and it was not significant for the strength of the adhesive joint.

Point 20: It is not clear why samples of 1.6 mm thickness with a different overlap were used? Why weren't 3mm samples with 12.5mm overlap used?
Response 20: For the materials prepared according to the EN 1465 standard, only cohesion failure mechanism were shown in area of Mg alloys. The changes in sample geometry (increasing the thickness of the material and reducing the length of the overlap) led to adhesive failure between the Mg alloy and the adhesive. This made it possible to evaluate the method of substrate preparing on the quality of the adhesive bonding. This information has been added to the manuscript.

Point 21: Line 259-262: What does irregularities on the surface mean?
Response 21: Surface irregularities were created during casting cutting. Due to the lack of pre-treatment by grinding or sandblasting, the scratches were not removed from the surface.

Point 22: Figure 11: Missing pictures g, h, I.
Response 22: Corrected in the manuscript. Thank you for the suggestion.

Round 2

Reviewer 2 Report

I am satisfied with the revision made. The work can be accepted now. 

Author Response

Dear Reviewer, 

Thank you for your opinion. 

Your sincerely,
Katarzyna Łyczkowska